

# First In Situ Measurements of the Prototype Tesseract Fluxgate Magnetometer on the ACES-II-Low Sounding Rocket

Kenton Greene[1], Scott R. Bounds[1], Robert M. Broadfoot[1], Connor Feltman[1], Samuel J. Hisel[1], Ryan M. Kraus[2], Amanda Lasko[1], Antonio Washington[1] and David M. Miles[1]

[1]Department of Physics and Astronomy, University of Iowa, Iowa City, IA, USA
[2]Peraton, NASA Sounding Rockets Operations Contract, Wallops Island, VA, USA

*Correspondence to*: Kenton Greene (kenton-greene@uiowa.edu)

**Abstract.** Ongoing innovation in next generation fluxgate magnetometry is important for enabling future investigations of space plasma, especially multi-spacecraft experimental studies of energy transport in the magnetosphere and the solar wind. Demonstrating the spaceflight capability of novel designs is an important step in the instrument development process; however, large-scale satellite missions are often unwilling to accept the risks of an instrument without flight heritage. The Tesseract - a novel fluxgate magnetometer sensor design-had an opportunity for an inaugural spaceflight demonstration on the ACES-II sounding rocket mission, which launched from Andøya Space Center in Andenes, Norway, in November 2022. Tesseract's design takes advantage of a new racetrack core geometry to create a sensor that addresses some of the issues that contribute to instability in more traditional ring core designs. Here we present the design of a prototype fluxgate magnetometer based on the new Tesseract sensor, its preflight characteristics, and an evaluation of its inflight performance aboard ACES-II. We find that the magnetic field measured by Tesseract over the course of the flight was in strong agreement with both the onboard ACES II reference ring core fluxgate magnetometer and the predictions of a geomagnetic field model. The Tesseract based magnetometer measured signatures of field aligned currents and potential Alfvén wave activity as it crossed an active auroral arc, we conclude that it performed as expected. Tesseract will be flown on the TRACERS Small Explorers (SMEX) satellite mission as part of the MAGIC technology demonstration currently scheduled to launch in 2025.

## 1 Introduction

Fluxgate magnetometers are important tools for spaceborne investigations of space plasma processes that are responsible for transporting energy and mass throughout the coupled Solar-Terrestrial system. Continued innovation in reliable, precise fluxgate magnetometer technology is important for enabling the next generation of space plasma science missions, especially multi-spacecraft investigations of magnetosphere and solar wind plasma. Recent multi-spacecraft missions, such as Swarm (Merayo et al., 2008), the Magnetospheric Multiscale Mission (MMS) (Russell et al., 2016), Space Technology 5 (ST-5) (Slavin et al., 2008), Themis (Auster et al., 2008), and Cluster (Balogh et al., 2001) have made important breakthroughs in our understanding of the multiscale plasma waves and current structures that drive the dynamic exchange of energy between the magnetosphere and ionosphere. For example, recent statistical studies of the morphology of terrestrial field-aligned current systems (i.e., Gjerloev et al., 2011; Lühr et al., 2015; Pakhotin et al., 2020; Cheng et al., 2021) depend on stable multipoint fluxgate measurements from missions like Cluster, Swarm and ST-5 to reliably resolve the small spatiotemporal magnetic fluctuations that enable precise monitoring of magnetospheric energy flux over variations in solar wind conditions. High stability, low noise multi-spacecraft fluxgate measurements have also made important contributions to our understanding of interplanetary plasma. For example, statistical studies of solar wind plasma, using fluxgate magnetometer data from MMS with a 20 pT/Hz instrumental noise floor (Russell et al., 2016), were able to resolve proton and electron inertial scale turbulence that plays an important role in the



heating and dissipation of solar wind and magnetosheath plasma (Chhiber et al., 2018; Chasapis et al., 2019). Recent multi-spacecraft missions have increasingly focused on flying three axis compensated fluxgate sensors in the interest of maximizing instrumental stability. However, three axis compensated sensor designs that have flown on these past missions have all been limited

to accommodate some variation of the traditional spiral wound ring geometry ferromagnetic core (e.g., Acuña et al., 1978).

New advances in fluxgate core technology (Miles et al. 2022; Narod and Miles 2023) are enabling new designs for fluxgate sensors not previously possible with ring cores. Miles et al., 2019 enabled the manufacturing of new miniature racetrack geometry cores which were found to have a more consistent yield and lower noise performance cores than the traditional ring core manufacturing process (Miles et al. 2022). Greene et al. (2022) developed a new Tesseract sensor design capable of accommodating

this new racetrack core geometry, while simultaneously addressing some of the design issues that are thought to cause instability in the more traditional ringcore design (i.e., Acuna et al., 1978; Wallis et al., 2015). Preliminary testing (Greene et al., 2022) found that the Tesseract sensor performs very well in metrics that are associated with instrumental stability and low noise and concluded that the sensor design looks promising for making low noise, stable magnetic measurements in a magnetospheric environment.

However, obtaining space flight heritage for new instrument designs is notoriously difficult. Large scale missions like

those described above are typically unwilling to accept the risks associated with an instrument that does not have demonstrated flight heritage. Sounding rockets are an excellent low-cost, low-consequence alternative for new instrument designs in need of an opportunity to demonstrate space flight capability.

The Tesseract fluxgate instrument was offered a flight opportunity on the ACES-II sounding rocket as a ride-along technology demonstration. ACES-II was a sounding rocket mission that used a high and low flyer pair to study the auroral electrical

current systems that are a key energy transport mechanism between the magnetosphere and the ionosphere. A Tesseract based fluxgate magnetometer prototype was launched aboard the ACES II low flyer sounding rocket from the Andøya Space Rocket Range in Andenes, Norway on November 20, 2022, at 17:21:40 UTC. The two-stage Black Brant XI Rocket reached apogee at an altitude of 188 km as it intercepted an active, discrete auroral arc. In this paper, we describe the design and construction of a prototype fluxgate magnetometer based on the Tesseract sensor design and present its in situ measurements of magnetic

perturbations associated with auroral electrodynamics.

## 2 The New Tesseract Based Magnetometer Design

Fluxgate magnetometers (Primdahl 1979) measure the static and low-frequency magnetic field by modulating the local magnetic flux and measuring the resulting electromotive force induced in a sense winding. A ferromagnetic core, periodically driven into magnetic saturation at frequency $f$. This effectively gates the local magnetic field, thereby inducing a $2f$ signal due to

the nonlinear magnetic permeability of the core as it enters magnetic saturation. The amplitude of this $2f$ signal is equal to the background magnetic field times a scale factor $S$. Two or more cores and windings, arranged orthogonally to one another, allow for the measurement of the full vector magnetic field.

The fidelity of a fluxgate's magnetic field measurement varies over time when its calibration parameters: sensitivity $S$, orthogonality $A$, and offset $O$, vary with changes in temperature or over time. For example, the alignment of a fluxgate's three

orthogonal axes, described by $A$, has been known to vary due to thermal and mechanical strain on the sensor (Primdahl 1979). Many spaceborne fluxgates, including the Tesseract, use global negative magnetic feedback to null the magnetic field inside the sensor which linearizes and extends the measurement range of the instrument (Primdahl and Jensen 1982). An inhomogeneous or inconsistent magnetic null is thought to contribute to instability of a fluxgate's instrumental offset (Ripka, 1992), orthogonality (Petrucha et al., 2015), and sensitivity (Korepanov and Marusenkov, 2012).





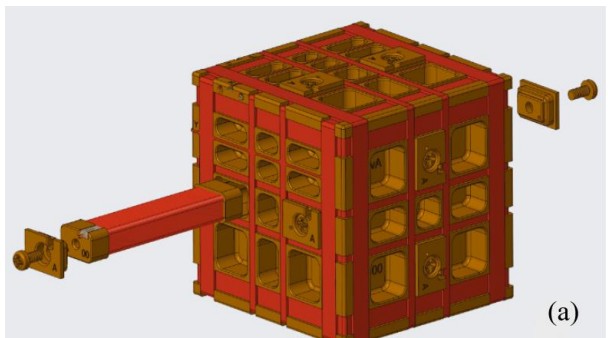
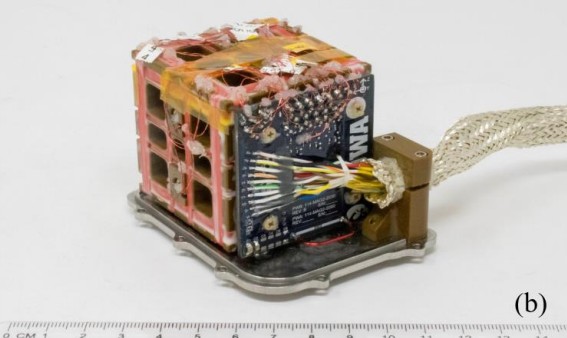

(a)  (b)

**Figure 1: (a) The Tesseract magnetometer design secures six miniature low noise racetrack cores within a symmetric block of 30% glass-filled Torlon engineering plastic. These racetrack cores, developed by Miles et al., 2022, are wrapped in a quasi-toroidal drive winding to modulate the permeability of the core and then covered in a solenoidal sense winding to sense the modulated signal. The Tesseract's feedback coils are wound on the same glass-filled Torlon base for structural stability. These feedback coils (red) are arranged in a three axis four-loop Merritt coil which creates a large region of magnetic homogeneity inside the sensor. (b) A photograph of the as-built Tesseract sensor that was flown on the ACES-II-Low sounding rocket.**

The Tesseract sensor (Greene et al., 2022) is a new design that takes advantage of a new racetrack core geometry to create a sensor that addresses issues described above that are thought to cause instrumental instability in more traditional fluxgate designs which use a ring geometry core (i.e Wallis et al., 2015; Miles et al., 2013). The Tesseract magnetometer design secures six racetrack cores within a symmetric block of 30% glass-filled Torlon (Figure 1a), which has a thermal coefficient of linear expansion similar to that of copper (~16 ppm/°C). Feedback coils are wound on the same symmetric base to reduce the tendency of the sensor's axes to skew with changes in temperature, potentially mitigating the effects of mismatched coefficients of thermal expansion which are thought to contribute to instrumental instability (Miles et al., 2017; Greene et al., 2023). These feedback coils (Figure 1a in red) are arranged in a four-loop Merritt coil which have been optimized to generate a large region of magnetic homogeneity in the vicinity of the cores in order to improve the reproducibility of the core's magnetization; thus, mitigating another potential source of instability. The sensor that was launched on ACES-II is shown in Figure 1b. An improved flight model with lower noise cores and optimized feedback electronics is currently being prepared for the Tandem Reconnection and Cusp Electrodynamics Reconnaissance Satellites (TRACERS) mission that will launch in 2025 as part of the MAGnetometers for Innovation and Capability (MAGIC) technology demonstration.

### 2.1 Tesseract Electronics

All the electronics are fit on a single 96 x 91x 21 mm board (Figure 2a) which is based on the analog design from the Cassope/ePOP fluxgate (Wallis et al., 2015) and has been modified to accommodate the Tesseract sensor. Figure 2b shows the major components of a single axis of the Tesseract electronics design. A resonant drive signal (I drive) is generated by a Field Programmable Gate Array (FPGA) and power amplified (PA). This signal is tuned using shunt capacitance and series inductance to pulse at a frequency of $f$ = 8.192 kHz (Figure 2c) and is sent into the quasi-toroidal drive windings of all six cores in series to periodically saturate the racetrack cores. The AC current output of the sense winding (I sense) is converted to a voltage using an op-amp based preamplifier (PRE). Figure 2d plots the voltage from the preamplifier when various background magnetic fields from -60000 nT to 60000 nT are applied to the sensor. The signal is then bandpass filtered in several stages (BP). The bandpass (plotted in Figure 2e) helps block the transformer coupled 1$f$ and 3$f$ drive signal and minimizes aliasing during digitisation.

This filtered signal then goes through a demodulation circuit (PSI) that inverts every half period of the 2$f$ signal to demodulate the fluxgate action while preserving polarity (Figure 2f). This is then filtered by a low pass filter (LPF) with a corner



frequency of 50 Hz resulting in a DC voltage that is directly proportional to the magnetic field. Finally, this voltage is digitized by a 20-bit analog-to-digital converter (ADC) at 16,384 samples per second and is then down sampled to 128 samples per second by

the FPGA for telemetry. Housekeeping data such as sensor and board temperature are digitized using a separate analog-to-digital converter (HK ADC). Finally, the data is transmitted to the rocket through a serial data interface, where it is timestamped.

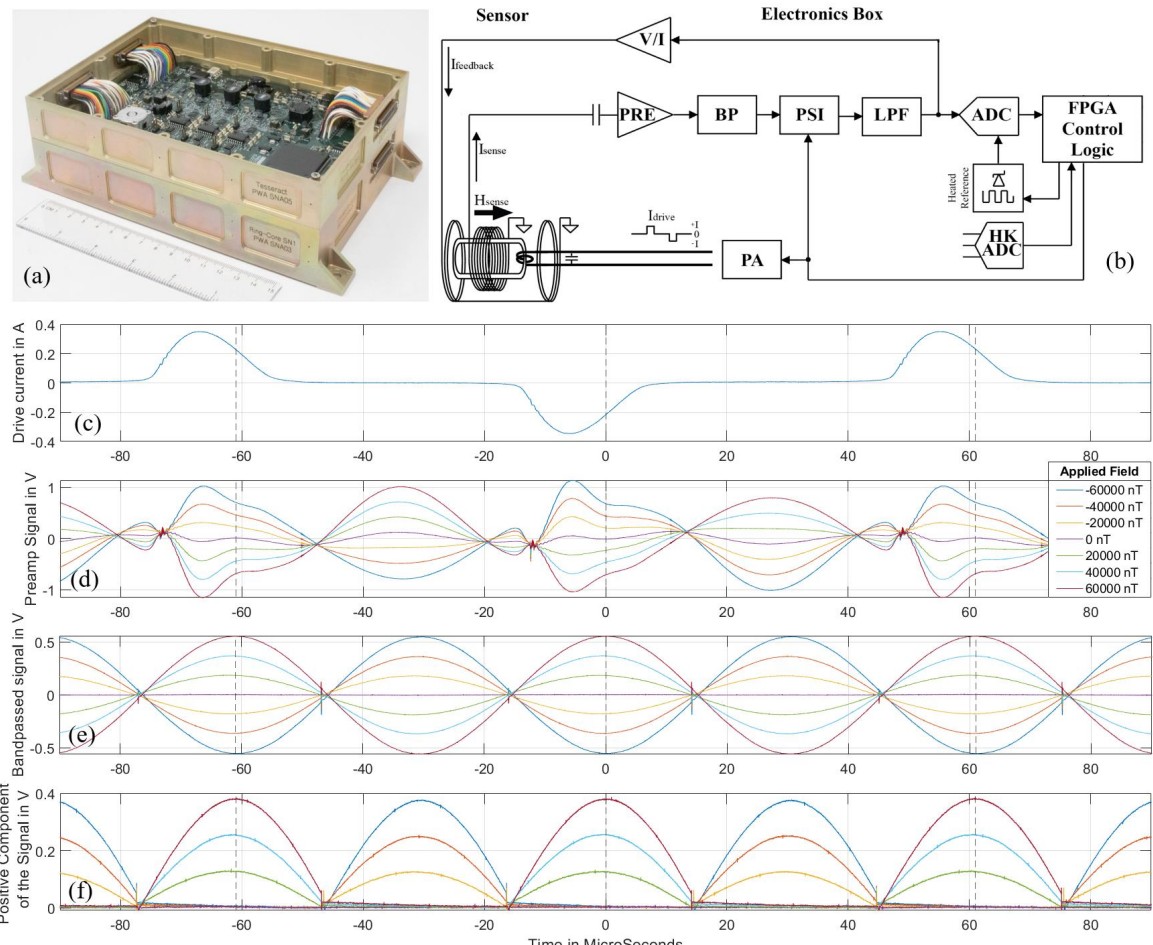

**Figure 2: (a) A photograph of the electronics board which was flown on ACES-II Low. The boards for two fluxgate instruments are stacked on top of one another. (b) A simplified schematic of Tesseract for a single instrument axis. (c) Drive pulses with a peak single-sided amplitude of about 350 mA, and a frequency of 8.19 kHz are applied to the sensor's drive windings. (d) The output of the sense**

**windings prior to filtering. (e) This signal is bandpass filtered to remove the 2f transformer coupled drive signal. The filtered signal is split into positive and negative voltage. (f) The negative component is inverted and added back to the positive component. Finally, this signal is lowpass filtered and the resulting DC voltage, which is directly proportional to the ambient magnetic field, is digitized at 128 samples per second.**

120         Output of the lowpass filter (LPF) is converted into an offset current (I feedback) by a transconductance amplifier (V/I) and fed as input to a feedback control loop. This current is sent into the Tesseract sensor's feedback coils to actively drive the field inside the sensor towards zero. The transconductance amplifier is intentionally unbalanced so that the voltage-to-current conversion factor depends on resistance of the feedback coils. This dependence on coil resistance is then tuned until the temperature effects of the coil resistance and the coil geometry are equal and opposite. This reduces the effect of temperature on the stability of the




instrumental sensitivity $S$ due to the temperature coefficient of linear expansion of the sensors coils (Acuna et al., 1978; Primdahl and Jensen 1982; Narod and Bennest 1990).

## 2.2 The ACES-II Low Magnetometer Payload

A prototype of the Tesseract instrument had an opportunity for a first flight demonstration on the ACES-II sounding rocket mission. ACES-II is a mission to study the auroral electrical current systems that are a key energy transport mechanism

between the magnetosphere and the ionosphere, particularly the distribution of Hall and Pedersen currents in the current closure region (i.e., Baumjohann 1983; Akari et al., 1989; Gjerlov and Hoffman 2000) and the balance of each in a stable auroral system. This investigation was carried out using two rocket payloads. A highflyer at an altitude of around 400 km observed the energy input from field aligned currents above the closure region, and a low flyer around the altitude of the closure current region measured the ionosphere's response to that input and the associated ionospheric energy dissipation.

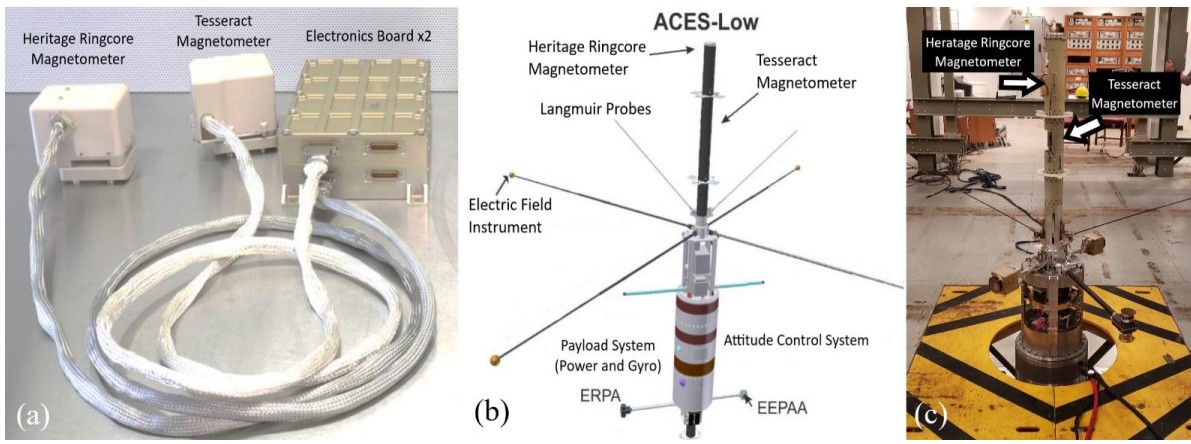

**Figure 3: (a) A photograph of the magnetometer instruments which flew on the ACES-II Low. The electronics boards for each sensor are stacked on top of one another. (b) A diagram of the ACES-II-Low science instrument payload. The Tesseract and Ringcore sensors are mounted in the rocket mast. The Tesseract is mounted inboard and the Ringcore is mounted outboard. (c) The Tesseract magnetometer and Ringcore magnetometer under test at the Wallops calibration facility while mounted on the rocket.**


Two fluxgate instruments flew on the ACES-II lowflyer sounding rocket (Figure 3a). The prototype Tesseract sensor was mounted inboard on the low flyer rocket mast, while a heritage ring core sensor design (Wallis et al., 2015) was mounted outboard (Figure 3b). The ACES-II payload was also equipped with an electric field instrument (Bonnell et al., 2008), Langmuir probes (Keltzing et al., 1998), as well as ion and electron top hat electrostatic analysers (EEPAA) (i.e. Calson et al., 1983) to measure the

pitch angle and energy distribution of auroral particles, and an Electron Retarding Potential Analyzer (ERPA) (Cohen et al., 2016) to measure thermal electrons, The rocket was equipped with an Attitude Control System (ACS) which utilized a gyro to allow for spin stabilization of the rocket to about 0.6 rotations per second and postflight determination of the attitude to less than 0.5 degrees. It was also equipped with an onboard Global Positioning System (GPS) which was used to acquire trajectory information.

## 3 Pre-flight Characterisation

A series of tests were conducted on the Tesseract magnetometer to quantify its performance prior to flight. The instrumental noise floor, sensitivity, orthogonality, offsets, and linearity were characterized at the University of Iowa Magnetometer Calibration Facility and again after integration with the rocket payload at Wallops Space Flight Facility (Figure 3c). The instrumental noise floor was also tested using a three-layered single axis shield facility at University of Iowa. These preflight





test data were used to assist with the calibration and de-spin of the in-situ data taken by Tesseract on ACES-II, which is described

in section 4.1.

| Parameter | Value |
|---|---|
| Sensor Mass | 343 g |
| Electronics Mass | 165 g |
| Sensor Dimensions | 50 x 50 x 50 mm |
| Electronics Dimensions | 96 x 91 x 21 mm |
| Power Consumption | 3000 mW |
| Sample Rate | 128 Sps |
| Magnetic Range | ±65,000 nT |
| Linearity from +/- 65000 nT* | 3 nT RMS |
| Sensitivity over Temperature‡ | 13 - 17 ppm/°C |
| Orthogonality over Temperature‡ | < 0.015 Degrees |
| Noise Floor @ 1 Hz | $15 - 21$ pT/√Hz |

\* After corrections in section 3.2 have been applied
‡ Between -45°C and 20°C (from Greene et al., 2022)

**Table 1: The characteristics of the early revision of the Tesseract fluxgate magnetometer that was flown on ACES-II Low sounding rocket. The temperature stability of the sensor base was characterized in a previous study and is described in Greene et al., 2022. The characterization testing procedures for noise and linearity are described in Section 3.**

**3.1 Pre-flight Calibration**

In each preflight calibration measurement, the Tesseract sensor was placed inside a large three-axis coil system. The coil system was used to generate a known ambient vector magnetic field $B_{Applied}$ of constant magnitude, which changed direction over time such that it sweeps out all solid angles of a sphere once every 5 minutes. The vector magnetic field measured by Tesseract $B_{Measured}$ was recorded over the course of this test. This data was used to calibrate the Tesseract magnetometer using a method

described in detail by Olsen et al. (2003) and Broadfoot et al. (2022), which exploits the relationship in Equation 1 to fit the instrument's intrinsic calibrations parameters - orthogonality $A$, sensitivity $S$, and offset $O$ - such that the vector residuals between the known vector field $B_{Applied}$ measured vector field $B_{Measured}$ is minimized.

$$B_{Applied} = R^{-1} A^{-1} S^{-1} (B_{Measured} - O) \qquad \text{(Equ 1)}$$


$A$ is a 3x3 matrix that describes the projection the magnetometer's three axes from a non-orthogonal frame onto an orthogonal frame (defined in Olsen et al., 2003). $S$ is a diagonal matrix with each diagonal element representing a scale value or sensitivity that converts voltage to nT for each axis. $R$ is a 3x3 rotation matrix consisting of three Euler angles that describe a rotation from the sensor frame into the frame of the rocket ACS. $O$ is simply the zero offset in each axis in nT.

An iteratively reweighted least squares linear regression (Holland and Welsch 1977) was used to estimate the best fit for the calibration parameters that minimizes $|B_{Measured} - B_{Applied}|$. The resulting calibration parameters for each axis are shown in Table 2. These calibration parameters were applied to the data taken over the course of the flight on ACES-II prior to the in situ de-spin and calibration which are described in Section 4.1.





**3.2 Instrument Linearity**


Equation 1 assumes that the sensor's response to the applied field is linear, thus characterization of the instrumental linearity is essential for the calibration to be effective. To characterize linearity, the coil system was used to generate a known ambient DC magnetic field. This field was ramped from -60,000 nT to 60,000 nT in a series of 4000 nT steps, and the field measured by the Tesseract magnetometer was recorded. Comparison of the known applied field with the measured field allowed us to characterize the instrumental linearity from -60,000 nT to 60,000 nT.


The Tesseract magnetometer uses negative magnetic feedback to null the magnetic field inside the sensor and extend the region of linear sensitivity of the permalloy core (Primdahl 1979). The efficiency of the magnetic nulling of Tesseract magnetometer prototype that was flown on ACES-II sounding rocket is lower than planned due to limitations in the development version of the feedback electronics which were not yet fully optimized for magnetic nulling due to flight campaign schedule constraints. At the time of integration with the ACES-II payload, the X axis feedback channel was in the process of being tweaked to maximize the magnetic feedback efficacy to extend the region of linearity to full earth field, the Y and Z feedback channels had not been optimized at all, thus, we expect to see a difference in the nonlinearity profile of the axes. The residuals of a robust multilinear regression fit (MATLAB Robustfit) to the field measured by the Tesseract against the known applied field are plotted on the left half of Figure 4 for each axis.


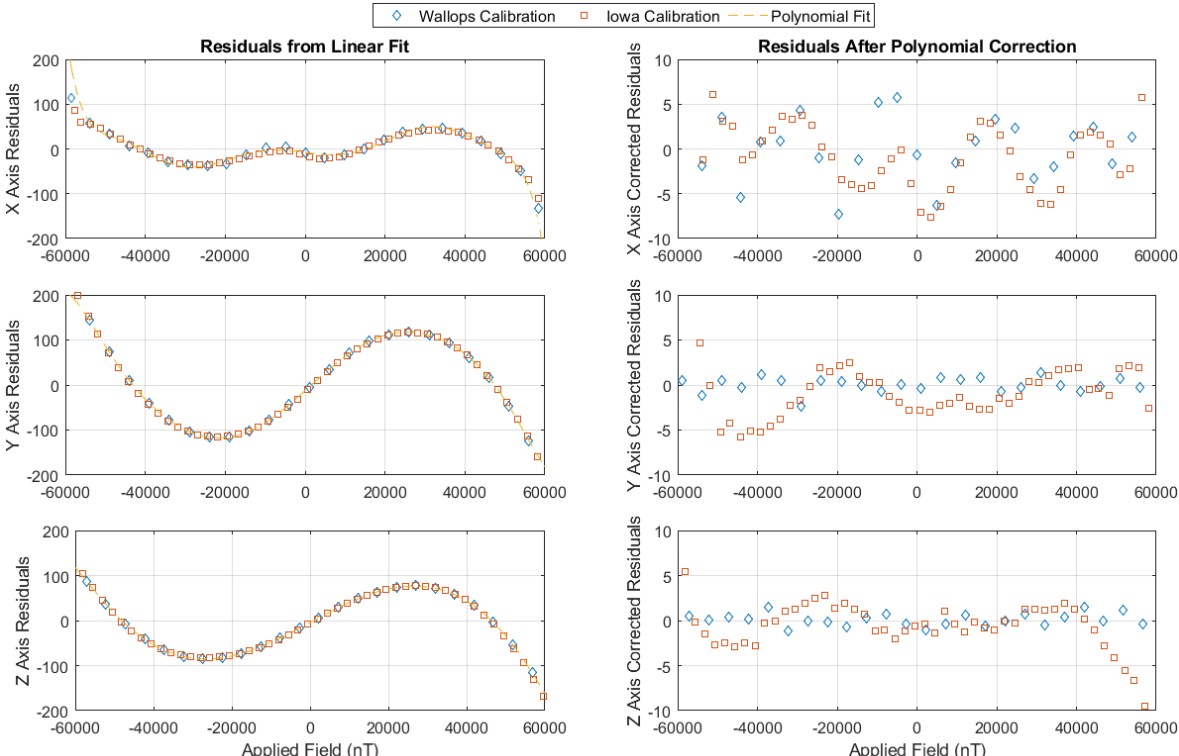

**Figure 4: The residuals of a linear fit to magnetic field measured by the Tesseract over the known applied field are plotted for each axis on the right-hand side. The residuals for the data taken using the Iowa calibration facility are plotted in red squares while the residuals for the same test using the Wallops calibration coils are plotted in blue diamonds. The polynomial fitted to the residuals is used to correct the flight data shown in yellow. The three agree to within 3 nT RMS over full magnetic range in each axis.**



These trends can be reliably described by a fifth order polynomial, a fit that represents the intrinsic nonlinearity of the prototype instrument. When the polynomial fitted to the residuals of the data taken during calibration at Iowa is subtracted from





residual of the data taken during the calibration at Wallops, the resulting difference between the two tests is ~3 nT RMS from -60000 nT to 60000 nT. This test was repeated three additional times at the University of Iowa magnetometer calibration facility. Measured nonlinearity from all tests agreed to within a few nanotesla despite different testing environments which gives us

confidence that this nonlinearity shown in Figure 4 is highly reproducible and can be reliably corrected for in flight. Subsequent iterations of the Tesseract's feedback electronics since the launch of ACES-II have been optimized for better linearity in preparation for a technology demonstration on the TRACERS small explorer satellites.

### 3.3 Instrumental Noise

The Tesseract sensor which flew on ACES-II was made from early versions of the racetrack design (Miles et al., 2019),

and the noise numbers are significantly lower in more recent fabrication efforts (i.e., Miles et al., 2022). The power spectral density noise floor of the Tesseract instrument that was flown on ACES-II was characterized prior to launch inside a single-axis four-layer mumetal magnetic shield (Figure 5a).

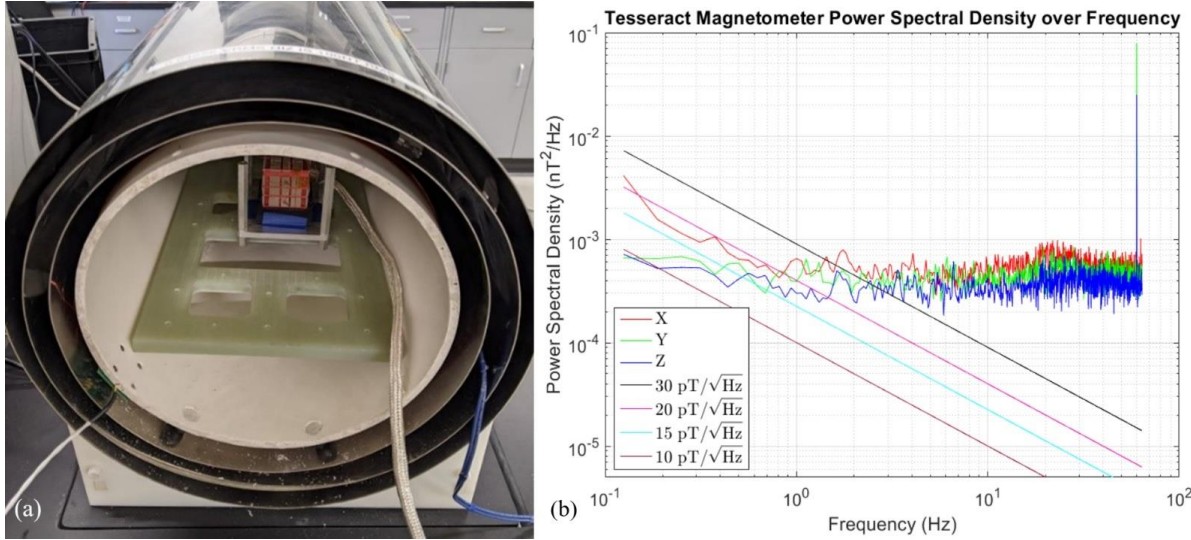

**Figure 5: (a) Tesseract's noise floor was characterized in a three-layer mumetal magnetic shield. (b) The power spectral density of the**
**instrumental noise plotted over frequency. The noise floor of the instrument ranges from 21 to 15 pT/√Hz at 1Hz for each of the three axes.**

To characterize the noise floor, twenty minutes of magnetically quiet data taken while the sensor was fixed in the magnetic shield. The power spectral density of this dataset was estimated using Welch's method of overlapping periodogram segments using

a Hanning window (MATLAB pwelch). The power spectral density is plotted against frequency for each axis in Figure 5b. The spectrally narrow spike at 60 Hz is due to the ambient residual magnetic signature of the laboratory. Robust linear regression (MATLAB robustfit) was used to fit a linear trend to the noise floor from 0.05 to 1.0 Hz, and this trend was evaluated at 1 Hz. The noise floor for each axis was found to be 21 pT/√Hz at 1Hz for the X axis, 19 pT/√Hz at 1Hz for the Y Axis, and 15 pT/√Hz at 1Hz for the Z Axis.

**4 In-Flight Performance**

The Tesseract magnetometer flew on the ACES-II low flyer: a two-stage Black Brant XI rocket which reached apogee at an altitude of 188 km. The ACES-II-Low sounding rocket was launched northward from the Andøya Space Rocket Range in Andenes, Norway, on November 20, 2022, at 17:21:40 UTC into an active auroral arc. The Tesseract Magnetometer in-flight data





was calibrated in post-processing against a reference geomagnetic field model during a magnetically quiet segment of data. The
Tesseract magnetometer's flight performance was evaluated via comparison to the heritage ringcore magnetometer and to the
reference geomagnetic field mode. Detailed science analysis of will follow in a subsequent publication. The preliminary results
shown here are to demonstrate instrument function.

### 4.1 Flight Calibration and Despin

Data taken by the Tesseract magnetometer over the course of the flight was de-spun and calibrated using a geomagnetic
field model as a reference. First, the polynomials that describe the preflight linearity, shown in Figure 4, were applied to each data
point to correct for instrumental nonlinearity. Then the rotation matrix (R) from the preflight characterization was applied to rotate
the sensor body frame of the instrument into the frame of the rocket ACS. The altitude, latitude, and longitude measured by the
Rocket GPS were used to evaluate the model vector field predicted by the CHAOS-7 magnetic field model (Finlay et al., 2020) at
every point over the course of the flight. The attitude solution was then used to rotate the CHAOS-7 model vector field into the
spinning frame of the rocket for direct comparison with the data.

| Calibration Parameters | X | Y | Z |
| --- | --- | --- | --- |
| Sensitivity (S) | 0.0122 nT/bit | 0.0114 nT/bit | 0.00120 nT/bit |
| Orthogonality (A) | 0.082° | 0.003° | 0.042 ° |
| Instrumental Zeros (O) | 9.3 nT | -14.8 nT | 13.1 nT |
| Rocket Offset | -87.3 nT | 430.3 nT | -328.0 nT |
| Rotation (R) | 0.14° | 0.22° | 0.49° |

**Table 2: The calibration parameters for the Tesseract magnetometer on the ACES-II sounding rocket. The four calibration parameters S, A, O, and R from Equation 1 are obtained from the preflight calibration testing. The rocket offset is fitted in-situ against a geomagnetic field model to account for the stray magnetic field of the rocket payload.**

Once the measured vector magnetic field and the model vector field were in the same rocket body frame, the intrinsic
calibration parameters S, A, and O, obtained from preflight testing (Table 2) were applied to the field vector measured by Tesseract
using Equation 1 to get the data in units of nT. Then an offset was fitted (Rocket Offset in Table 2) to the CHAOS-7 model field
in all three axes during a quiet segment of data immediately before the science region (17:24:00 to 17:24:30 UTC). We attribute
this offset to the stray fields of the rocket motors below the payload section that were not present during preflight calibrations. The
background model field was subtracted from the measured magnetic field in the spinning rocket frame. Finally, the attitude data
from the ACS was used to rotate the data from the spinning rocket frame into an East-North-Up geophysical coordinate system.

### 4.2 Comparison with Reference Ring Core Magnetometer and Geomagnetic Field Model

The same process described above was used to de-spin and calibrate the heritage ring core geometry sensor. Once
calibrated and rotated into the same geophysical coordinate system, the fields measured by the Tesseract fluxgate and the heritage
ringcore magnetometer could be directly compared. Figure 6 plots the field measured by Tesseract around the time of apogee of
the rocket trajectory in red in the Eastward (a), Northward (b), and Upward (c) directions as well as the magnitude (d). The field
measured by the ring core sensor in each direction is plotted alongside it in blue.



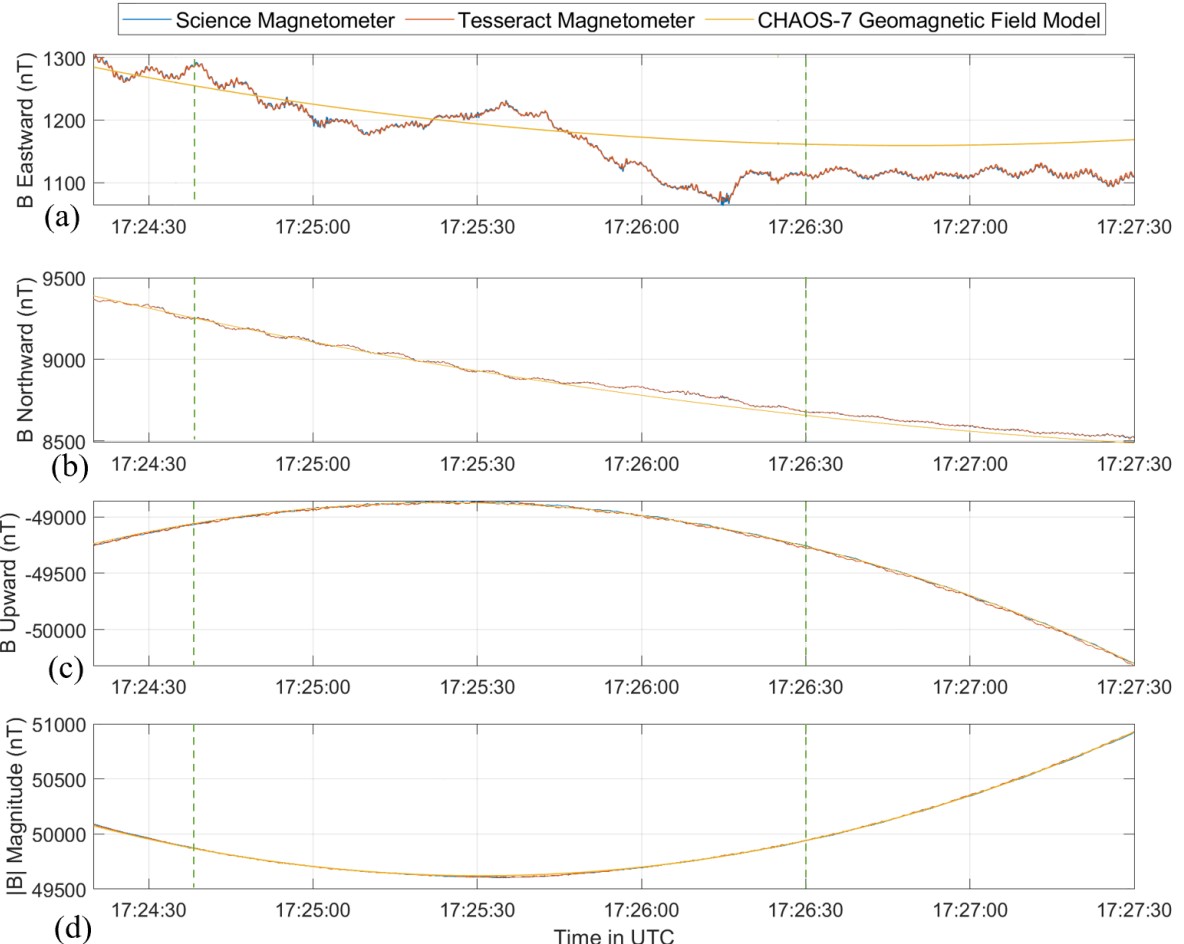

**Figure 6: During a quiet segment of data, between 17:24:00 and 17:24:30, the Ringcore and Tesseract data was calibrated using the calibration values obtained during preflight testing. The magnetic field measured by the Tesseract magnetometer is plotted over time in red, and the measured field by the ring core is plotted in blue for the Eastward (a), Northward (b) and Upward (c) directions. The scalar magnetic field is plotted in (d). The magnetic field predicted by the CHAOS 7 geomagnetic field model was evaluated at the rocket's location over the course of the flight and is plotted in yellow. The region where we expect to observe auroral activity based on the All-sky image is bounded by green dotted lines.**

Tesseract exhibits very good agreement with the heritage ringcore magnetometer to within 1.91 nT RMS in the Eastward direction (Figure 6a), 3.08 nT RMS in the Northward direction (Figure 6b), and 5.48 nT RMS in the Upward direction (Figure 6c), and 0.62 nT in magnitude (Figure 6d) over this time range. These small discrepancies and remaining periodicity are likely due to small (>0.05°) rotation errors between the sensor coordinates system and the ACS coordinate system that was not accounted for in the preflight calibration.

The longitude, latitude, and altitude from the attitude data were fed as inputs into the CHAOS-7 geomagnetic field model and were used to model the Geomagnetic field in the Eastward, Northward, and Upward directions. The measured field and the model field agree to within about 25 nT RMS in each axis except from 17:24:40 to 17:26:30, which correspond to the time that the ACES-II payload traversed an auroral arc based on comparison between rocket GPS data and the data from an all-sky imager taken in Skiboten, Norway (Figure 7). We take this agreement with the science magnetometer and the CHAOS-7 magnetic field model as validation that the Tesseract instrument functioned as expected.





## 5 Measurement of Magnetic Signatures Associated with Auroral Currents

A map of the ACES-II Lowflyer trajectory measured by the ACS GPS is shown in Figure 7 plotted on top of an All Sky
280  image of 630 nm wavelength light taken at the Tromso Geophysical Observatory in Skiboten, Norway, at 17:25:00 UTC, which
has been projected onto the corresponding longitude and latitude. Based on the trajectory an All Sky Imager data, ACES-II is
expected to have traversed at least one active auroral arc between 17:24:40 and 17:26:30 UTC. The visible auroral arc remained
relatively stable over the course of the flight, but was slowly drifted equatorward. However, at about 17:20:35 UTC, eight minutes
after the flight ended, a surge drove the visible arc southward, and it evolved quickly into a dynamic auroral substorm. The arc was
285  on the horizon so any fine structure would have been obscured.

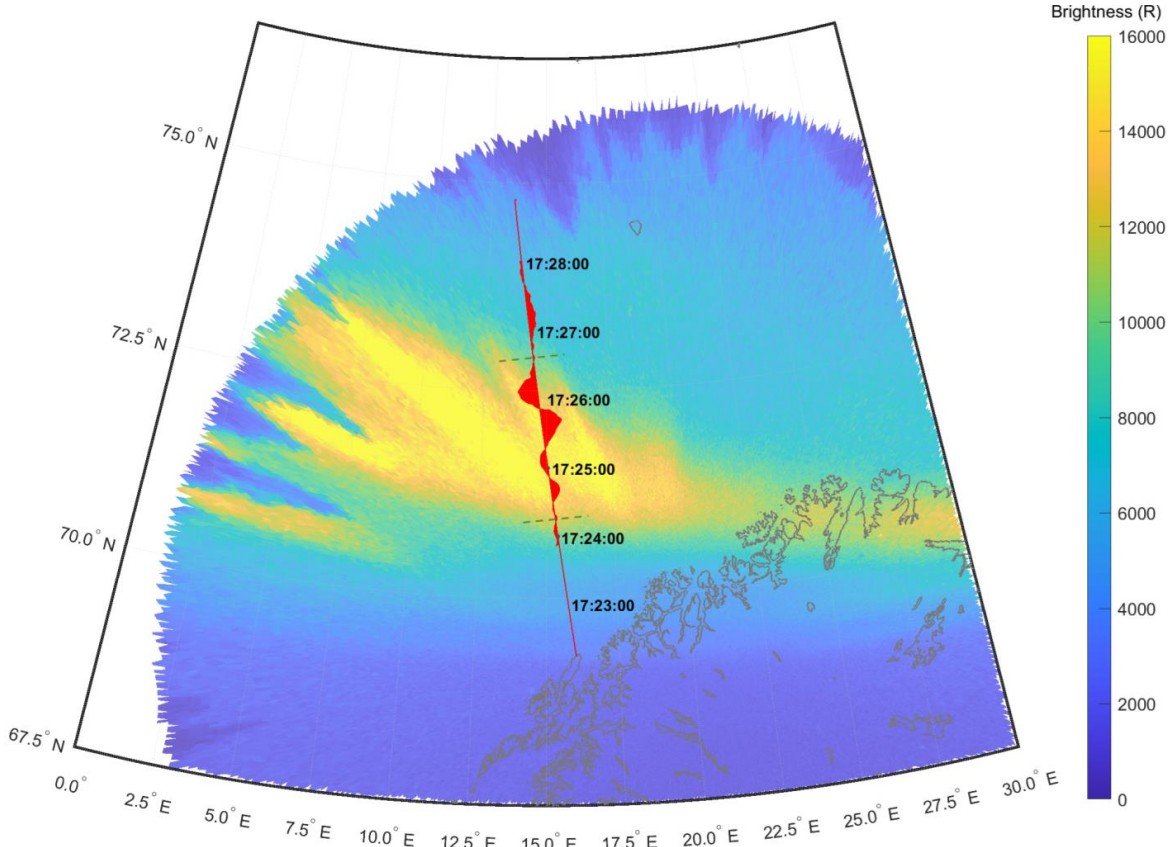

**Figure 7: An All-Sky image taken in Skiboten, Norway at 17:25:30 is projected onto its approximate location and is plotted alongside
the trajectory of the ACES-II lowflyer as measured by the rocket GPS. The eastward DC magnetic field that Tesseract magnetometer
measures over the course of the flight is plotted in red. Around the apogee, from 17:24:40 UTC to 17:26:30 UTC, the ACES-II low flyer
passes through at least one an active auroral arc.**

290

The deviation of the magnetic field measured by Tesseract from the background model field over the course of the flight
is plotted in red. The measured field deviates from the nominal magnetic field from 17:24:40– 17:26:30 by a maximum of about
50 nT in the East direction, 25 nT in the North direction, and 15 nT in the Up direction. This time range corresponds to the time
that the ACES-II payload traversed an auroral arc based on data from an All-Sky Imager (Figure 7). The magnetic field in the
Eastward direction, implying that these magnetic fluctuations are primarily measurements of the east-west field aligned current
sheet that is likely associated with the aurora.



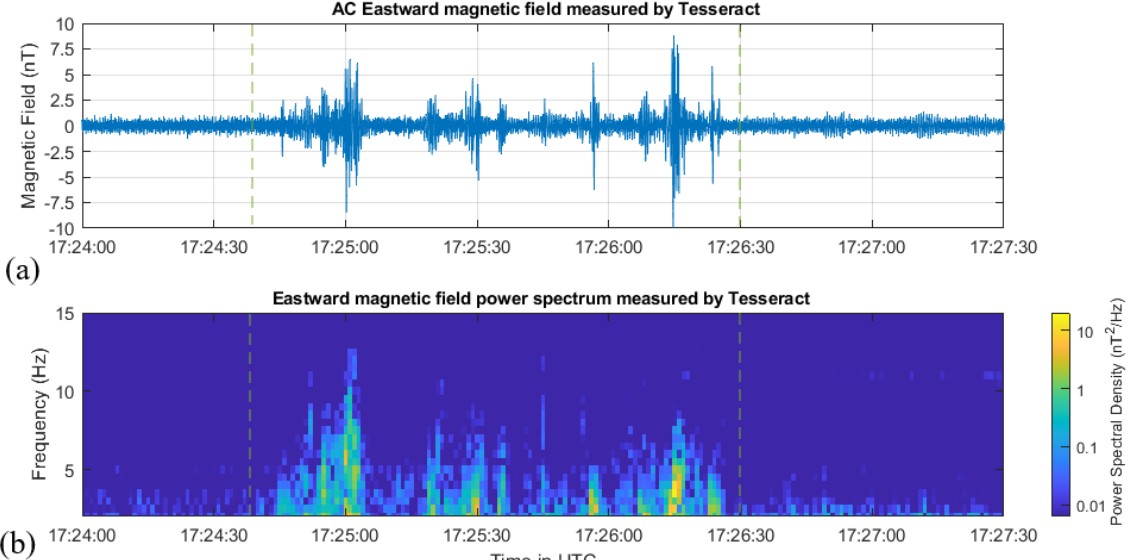

**Figure 8: Highpass filtering the data at 1 Hz removes the DC background magnetic field, allowing us to highlight the AC signature in the data. (a) We observe magnetic fluctuations with amplitudes up to 8 nT at the same times that the ACES-II Low payload crosses the visible auroral arc show in Figure 7. (b) The power spectrum of this data shows that these fluctuations have frequencies up to 8 Hz.**


Figure 8a plots one of the transverse components of the magnetic field (Eastward) measured by Tesseract, which has been computationally highpass filtered below 1 Hz using a simple running mean filter (MATLAB highpass) to suppress the background magnetic field. This crudely filters out the residual spin tone leaving just the higher frequency fluctuations to highlight the AC

magnetic field (Figure 8a). The spectrogram of this data, plotted in Figure 8b shows, mostly broadband magnetic fluctuations from about 1 – 8 Hz.

The magnetic perturbations encountered from 17:24:40 to 17:26:30 UTC (marked by dotted green lines) could potentially be signatures of Alfvén waves which are responsible for transferring energy between the magnetosphere and ionosphere. Alfvén waves with broadband frequencies up to 8 Hz are oftentimes observed in conjunction with the discrete aurora (Stasiewicz et al.,

2000 and the references therein) and can be responsible for accelerating electrons that power the aurora (Schroeder et al., 2021; Kletzing 1994; Chaston et al., 2007). Alfvén waves are also thought to play a role in driving Joule heating (Hartinger et al., 2015; Chaston et al., 2006) and ion outflow (Fernandes et al., 2016) in the ionosphere. A more detailed scientific investigation using multiple instruments on ACES-II is underway and will be published separately.

**6 Summary and Conclusion**

A new fluxgate magnetometer design is presented. The procedures and results of a full calibration and characterization of the instrument's preflight performance are presented. In situ performance of the instrument's first space flight was found to be in good agreement with the preflight calibrations, the onboard heritage science magnetometer, and the CHAOS-7 geomagnetic field model. We find that the magnetic field measured by the Tesseract magnetometer corresponded with the heritage science

magnetometer to within 5.5 nT RMS in all three axes, and with the field predicted by CHAOS-7 geomagnetic field model to within 25 nT RMS during geomagnetically quiet times. We conclude that the Tesseract magnetometer performed as expected, and the calibration efforts were successful.



Additionally, the Tesseract magnetometer observes perturbations in the background magnetic field as large as 50 nT and AC magnetic fluctuations between 1 and 8 Hz which are coincident with the crossing of an active auroral current sheet, suggesting that these fluctuations may be involved with transporting energy and accelerating auroral electrons and ions that couple the ionosphere to the magnetosphere. We demonstrate the capability of a new magnetometer design to measure a geophysical magnetic field in a space environment that is potentially relevant for scientific studies of auroral electrodynamics.

### 6.1 Future Work

Once the data for the rest of the ACES-II instruments has been processed and calibrated, we will compare the data taken by Tesseract with the data taken by the onboard electric field instruments to determine whether the observed magnetic fluctuations are signatures of Alfvén waves or quasi static currents that have been doppler shifted in the rocket frame (Knudsen et al., 1992). We will compare these data to the data taken with the electron and ion instruments to determine whether these fluctuations are associated with electron or ion acceleration and transport and whether they might play a role in transporting energy from the magnetosphere to the ionosphere. We will also compare this to the plasma density measurement taken with the onboard Langmuir probes which will be used to estimate the Alfvén speed and ionospheric conductivity.

The development of the Tesseract magnetometer is ongoing. The closed loop feedback electronics are being optimized to minimize nonlinearity on future iterations of the instrument. The Tesseract magnetometer will be flight demonstrated again with lower noise racetrack cores (Miles et al., 2022) and improved negative feedback electronics on the upcoming Tandem Reconnection and Cusp Electrodynamics Reconnaissance Satellites (TRACERS) Small Explorer mission (Kletzing, 2019) as part of the MAGIC technology demonstration.

### Code and Data Availability

Data and source code used in the creation of this paper can be accessed by contacting the authors.

### Author Contributions

K. Greene led the Tesseract sensor development, calibration, and data analysis as the Tesseract instrument PI and wrote the manuscript with contributions from all authors. D. M. Miles provided oversight as the ACES-II magnetometer instrument PI, assisted with the interpretation of the data, and created the visualization software that generated the render of the flight trajectory and Keogram overlay. S. R. Bounds oversaw construction, integration, and testing of the ACES-II science payload and led the ACES-II sounding rocket campaign as the mission's principal investigator, providing funding and the flight demonstration opportunity for Tesseract. R. Broadfoot led the in-situ calibration efforts and assisted with preflight calibration testing. C. Feltman assisted with the preflight calibration and in-situ de-spin efforts. S. J. Hisel led the design, assembly, and optimization of the new fluxgate electronics package to accommodate the Tesseract sensor. R. M. Kraus oversaw the construction of the attitude solution from the raw attitude control system data and synchronized it with the telemetry to support the inflight de-spin. A. Lasko oversaw the instrument fabrication, integration, and calibration efforts as the ACES-II magnetometer instrument project manager. A. Washington programmed the FPGA, ensured synchronization of the Tesseract instrument with the rocket telemetry, and developed software to assist with calibration, data visualization, and data processing.

### Competing Interests

The Authors declare that they have no competing interests.



**Acknowledgments**

This work is only possible because of the effort and professionalism of the entire ACES II team which made the ACES-II rocket campaign a success. The authors would like to extend a heartfelt thank you to manager Jay Scott and the entire NSROC payload team: Valarie Snell, Darren Ryan, Zach Schulze, and Eric Pittman for their work with telemetry and power, as well as Graham Taylor, Mike Johnson, Mitch McPhial, Gary Snead, Brian Brittingham, and Beth West for their technical expertise.

The authors would specifically like to thank Jeff Dolan for his expertise and support over the course of the ACES-II sounding rocket campaign. The authors would like to thank Matthew G. Finley for his hard work supporting integration and mag cal. The authors would also like to thank Michael Webb for his excellent work with electronics board fabrication and Katherine Deasy for her work in configuration management.

Early development activities and much of the research infrastructure used in this work was supported by faculty startup funding for David Miles from the University of Iowa.

This material is based upon work supported by the National Aeronautics and Space Administration under Grant No. 80NSSC19K0491 issued through the Science Mission Directorate and Contract No. 80GSFC18C0008 administered by Goddard Space Flight Center.

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
