# Peer review of "First In Situ Measurements of the Prototype Tesseract Fluxgate Magnetometer on the ACES-II-Low Sounding Rocket"

_EGUsphere, 2024_

## Author Comment (AC1)

Response to comments on 'First In Situ Measurements of the Prototype Tesseract Fluxgate Magnetometer on the ACES-II Low Sounding Rocket' by reviewer #1 on February 5th, 2024

We thank the referee for the constructive comments which we have incorporated into the manuscript. The reviewer raised an important issue about the temperature dependance of the instrumental sensitivity as well as other corrections, which we address below. Referee comments are in plain text our responses in *italics* and any content added to or changed in the manuscript are in "*quoted italics*"
* * *
Temperature stability is an important factor of magnetometer operation, especially in the case of spacecraft on-board installation. Declared sensitivity temperature dependence of 13-17 ppm/deg is exactly a thermal expansion coefficient of feedback coils. Is it wholly satisfactory for this mission? There is no description of how this figure was measured. Moreover, temperature behavior of polynomial coefficients for non-linearity correction (which seem to be not dependent on feedback coils), has not been addressed at all. Your consideration on the subject would be relevant and instructive.

*The reviewer highlights the importance of the dependance of sensitivity on temperature for fluxgate measurements and raises important questions about the characterization of Tesseract's sensitivity over temperature. The reviewer points out that temperature dependance of Tesseract's sensitivity is shown as 13-17 ppm/deg in Table 1. This figure was measured in testing detailed in a previous study (Greene et al., 2022) which found the temperature stability of the Tesseract's base and feedback coil without any dependence on cores and electronics. This was accomplished by temporarily configuring the sensor's feedback windings as an air-core search coil magnetometer. The sensor was placed in a thermally insulated box made from 10 cm-thick polystyrene to create a controlled temperature environment for the sensor. The polystyrene box is then placed within the Merritt coil system and the coil system is used to generate a known 60,000 nT AC field in each axis. Dry ice is placed inside the box to chill the sensor, and measurements are taken after the dry ice has sublimated and the sensor is slowly warming. A platinum RTD temperature sensor is attached to the sensor and records the change in temperature as the sensor returns to room temperature (Fig. 10). As the Tesseract sensor temperature slowly increased, the voltage induced in the Tesseract prototype sensor's feedback windings and the RTD were measured. The details of the test and measurements are described in depth in Greene et al., 2022.*

*We have added the following text in section 3 to clarify the origin of these measured values:*

*A sentence has been added to Line 160: "Table 1 shows the characteristics of the Tesseract Sensor which flew and ACES-II.  The temperature stability of the Tesseract sensor's base and*

*feedback coils, without any dependence on cores and electronics, were characterized in a previous study (Greene et al., 2022)."*

*Line 158 now reads "The temperature stability of the Tesseract sensor's base's sensitivity and orthogonality was characterized in a previous study and is described in detail in Greene et al., 2022."*

*The also reviewer points out the importance of temperature dependence of a fluxgates characteristics such as sensitivity and nonlinearity when feedback electronics are used to null the field around the fluxgate cores. This characterization test was not performed on Tesseract flight model before the ACES-II rocket flight. The expected change in temperature over the course of the 10-minute rocket flight was expected to be minimal. The measured sensor temperature ended up changing by only about 4 degrees °C. This small change in sensor temperature, we expect, will have a very small effect on calibration and nonlinearity and that other sources of error in the calibration such as uncertainties in rotation angles and the offset due to the stray field of the rocket payload will dominate. An in-depth characterization of the temperature dependence of the Tesseract instrument's calibration is being carried out in preparation future missions, such as the upcoming TRACERS satellite mission where temperature changes as large as 70 °C will have a more significant impact on calibration.*

*We have added this context in line 164: "A full thermal calibration of entire Tesseract instrument is not explored in this paper. The sensor temperature changed by only 4 °C over the course of the flight aboard ACES-II, so the errors in calibration introduced from changes in temperature are expected to be minimal. An complete temperature calibration of the Tesseract instrument which includes the cores and electronics will be performed in preparation for the upcoming TRACERS SMEX satellite mission."*

Line 245: Sensitivity figure for Z direction seems to be erroneous.

*We thank the reviewer for their careful eye in catching this error. The Z direction has an erroneously added zero. We have corrected this mistake in Table 2.*

*Please find that the following changes have been made as suggested:*

Line 35: "20 pT/Hz" should be 20 pT/sqrtHz

Line 211: "inside a single-axis four-layer mumetal magnetic shield (Figure 5a)" According to Fig.5 and its legend, it is "a three-layer mumetal magnetic shield".

Line 223: "Robust linear regression was used to fit a linear trend to the noise floor from 0.05 to 1.0 Hz, and this trend was evaluated at 1 Hz" Consider "Robust linear regression was used to fit a linear trend from 0.05 to 1.0 Hz, and the noise floor was evaluated at 1 Hz".

Line 302: "computationally highpass filtered below 1 Hz". Should it be "over 1 Hz"?

---

## Author Comment (AC2)

Response to comments on 'First In Situ Measurements of the Prototype Tesseract Fluxgate Magnetometer on the ACES-II Low Sounding Rocket' by reviewer #2 on February 15th 2024

We thank the referee for the constructive comments which we have incorporated into the manuscript. The reviewer raised important issues about the comparison between the heritage science ring core magnetometer and the Tesseract magnetometer as well as other corrections, which we address below. Referee comments are in plain text our responses in italics and any content added to or changed in the manuscript are in "quoted italics"
* * *
The authors present the design of a prototype fluxgate magnetometer based on its pre-flight characteristics and an evaluation of its performance during a short flight aboard sounding rocket ACES-II. The paper is well written, understandable and an appropriate number of citations is included. The ability of the new magnetometer to perform geophysical magnetic field measurements in the space environment is clearly demonstrated.

General comments:

1. The primary (science) magnetometer, which is the main reference for the in-flight comparison, should be briefly discussed and relevant literature should be cited.

*We agree with the reviewer's assessment that a discussion of the science magnetometer should be included. The ring core sensor used in this paper is based a heritage design for a spaceborne fluxgate magnetometer first developed by Acuña et al., 1978 which used 1" diameter S1000 ringcores from Infinetics. The design is nearly identical to the sensors described by Miles et al., 2013 and Wallis et al., 2015.*

*The following context about the design of the heritage ring core science magnetometer, along with the relevant citations, have been added to Line 143 where the ringcore sensor is introduced, Line 143 now reads: "The ring core sensor's design has its heritage in the NASA MAGSAT (Acuña et al., 1978) which uses two 1" diameter ring cores which are each wound with two orthogonal solenoidal coils, providing two measurements in the plane of each ring. The design is nearly identical to the sensors described by Miles et al., 2013 and Wallis et al., 2015."*

*And in Line 260 the text now reads: "The same process described above was used to de-spin and calibrate the heritage ring core geometry sensor, which uses the same design described in Miles et al., 2013."*

2. The pre-flight calibration of the three Euler angles of the rotation matrix and its accuracy should be discussed, as it is assumed to be the main cause of the difference between the prototype and the primary (science) magnetometer.

*The reviewer highlights the importance of the rotation in the calibration, especially since it is suspected that it may be a possible contributor to uncertainty of our calibration. We agree that this is important information to include, and the following context has been added to clarify on line 81:*

*"R is a 3x3 rotation matrix consisting of three Euler angles that describe a rotation from the sensor frame into the frame of the rocket ACS.  Uncertainty in the measurement of the Euler angles is dependent ability to accurately align the ACS with the coil system during calibration. We estimate that this alignment is good for angles larger than 0.05 degrees."*

3.  It is not clear for how long the magnetometer was actively measuring.

*We agree with the reviewer that the length of time that the instrument was measuring should be stated explicitly in the body text. We have added this information on line 240 which now reads: "The Tesseract Magnetometer took measurements of the ambient magnetic field over the course of the flight from launch, until 17:28:50 UTC when connection to the rocket was lost upon reentry."*

4.  It is said that a quiet period between 17:24:00 and 17:24:30 was used for the in-flight calibration, but not the entire 30 seconds are shown in Figure 6. What do the data look like before the "quiet period"?

*The reviewer points out that Figure 6 does not show the entire quiet period and cuts off the plot too early. We thank the reviewer for bringing this error to our attention. The new Figure 6 in the revised manuscript plots the magnetic field starting at 17:24:00 UTC and shows the complete quiet area where the instrument was calibrated from beginning to end. The new Figure 6 is shown below and has also been incorporated into the revised version of the manuscript:*

[Figure]

5. The measured and the modelled field obviously agree to within 25 nT RMS outside of the scientifically interesting period. What is the standard deviation of the difference within the mentioned crossing of the active auroral arc? The plots in Figure 6 do not indicate a big difference between the two phases.

*We agree with the reviewer that including the RMS deviation between the measured and model field in the active auroral region is a figure that is useful to include for comparison and for completeness. We have added this information in a sentence on line 290: "In the region associated with the auroral arc the measured field and model field agree within 37 nT RMS in each axis."*

6. The performance discussion would benefit from plotting the difference between the prototype and science magnetometer to deepen the demonstration of the good match. The mentioned alignment mismatch between the two sensors could be calibrated based on the flight data which would further reduce the reported RMS deviation.

*We agree with the reviewer's assessment that plotting the difference between the field measured by Tesseract and the field measured by the Ring core science magnetometer would be*

*illustrative in demonstrating the agreement between the sensors and help to reinforce the main result of the paper: that the Tesseract magnetometer performed as expected as a functioning magnetometer over the course of the flight. We have added a new Figure 7 which plots the difference between the sensor's measurements in each axis on line 286:*

[Figure]

*We have also added a figure caption on line 287 which reads: "Figure 7: The difference between the magnetic field measured by the heritage ring core science magnetometer and the magnetic field measured by the Tesseract is plotted for the Eastward (a) Northward (b) and Upward (c) directions along with the scalar (d) field. The region where the rocket payload is expected to have crossed the auroral arc is bounded by dashed green lines."*

*Changes to the numbering of the subsequent figures have also been made accordingly in the revised manuscript.*

7. A comparison of the filtered data from both magnetometers would show that also the actual science event was measured correctly by the prototype sensor.

*We agree with the reviewer that a plot showing the agreement of the magnetic field measured by Tesseract and the magnetic field measured by the science magnetometer in the science region would demonstrate that the science region was measured correctly by the prototype. A Figure 7 has been added, which shows the difference between the field measured by the Tesseract and the field measured by the ring core. The region of data bounded by the green dashed lines shows the difference of the fields measured by the two sensors in during the science event.*

*A sentence is also added which quantifies the agreement between the two sensors in the science event region:*

*Line 283 now reads: "The Tesseract and Ring core measured the same field in the region of auroral activity (bounded by green dashed lines in Figure) to within 5.53 nT RMS in all three axes."*

Specific comments:

*The following changes have been made as suggested in the revised version of the text:*

Line 35: The instrumental noise of the MMS sensors in low range is less than 8 pT/sqrt(Hz).

Line 146: … to measure thermal electrons.

Line 231: … detailed science analysis of it will …

---

## Author Response (AR2)

Response to Final Peer Review Referee Comments #2

We thank the reviewer for their constructive comments which we have incorporated into the manuscript. The Reviewers comments are in plain text. The Authors responses are in "quotes" and the changes made in the text of the revised manuscript are in "*quoted italics*".
* * *
2. The pre-flight calibration of the three Euler angles of the rotation matrix and its accuracy should be discussed, as it is assumed to be the main cause of the difference between the prototype and the primary (science) magnetometer.

"The reviewer highlights the importance of the rotation in the calibration, especially since it is suspected that it may be a possible contributor to uncertainty of our calibration. We agree that this is important information to include, and the following context has been added to clarify on line 181:"

"*$R$ is a 3x3 rotation matrix consisting of three Euler angles that describe a rotation from the sensor frame into the frame of the rocket ACS. Uncertainty in the measurement of the Euler angles is dependent ability to accurately align the ACS with the coil system during calibration. We estimate that this alignment is good for angles larger than 0.05 degrees.*"

Reviewer Comment 2a: It is not clear what exactly is meant with the statement "We estimate that this alignment is good for angles larger than 0.05 degrees."? It was asked for the accuracy of the pre-flight calibration of the Euler angles.

2a. "The reviewer points out the ambiguity in the text describing the accuracy of the characterization of the Euler angles during preflight calibration. To clarify the uncertainty of the Euler angles and its origins, we have added in the following text to clarify in Line 181 which now reads:"

"*Uncertainty in the measurement of the Euler angles is dependent on the ability to accurately align the ACS with the coil system during calibration. We estimate that this alignment is good for angles larger than 0.05 degrees and thus the uncertainty in the Euler angles measured during preflight calibration is 0.05 degrees.*"

Reviewer Comment 2b: It is appreciated that the new Figure 7 shows the differences between the Tesseract and the science magnetometer, but the authors should give an indication of what is causing the rather large RMS value of 5.53 nT.

2b. "The reviewer highlights the importance of clearly stating the causes of the difference in measured field between the two magnetometers. The following text has been changed on Lines 285-286 to clarify the causes of the RMS difference in measurement between the two sensors*:*

"*These discrepancies and remaining periodicity are due to residual spin tones introduced by small (>0.05°) uncertainties in the alignment between the sensor coordinate system and the ACS coordinate system that were not accounted for in the preflight calibration. The preflight calibration procedure described in Section 3.1 was only able to measure Euler angles larger than 0.05°. The uncertainty in the*

*preflight characterization of the Euler angles reduces the ability to effectivity remove the effects of the rocket's spin through Earth's magnetic field, leading to a residual spin tone between the two sensors (Figure 7)."*